# Depression among older parents' and their associated factors after adult children's migration: A mixed method study in an urban setting of Kathmandu Metropolitan City

Soniya Shrestha[1,2]*, Bigya Tuladhar[3], Yamuna Chhetri[2], Sapana Ghimire[2], Madhusudan Subedi[2]

**1** Ministry of Health and Population, Kathmandu, Nepal, **2** School of Public Health, Patan Academy of Health Sciences, Lagankhel, Lalitpur, Nepal, **3** Engineering and Development Department, NOVA Dynamic Media Company Limited, Tsuen Wan, Hong Kong

* soniashrestha10@gmail.com

## Abstract

With increasing life-expectancy, the older population of Nepal has also been increasing with Nepal census 2021 reporting 60 years and above population to be 10.21%. Nepal also has a high migration rate with 2.2 million absentee population reported in the 2021 census. Outmigration of adult children can negatively impact parental health, including loneliness, depression, and poor psychological health, despite potentially increasing parents' financial resources and access to welfare services. This study aims to evaluate the level of depression and their associated factors among the empty nest older population residing in ward number 9 of Kathmandu Metropolitan City after adult children's migration. A cross-sectional mixed-method study was carried out among 140 older parents aged over 60, with all children migrated abroad, selected randomly from a social security list. Depression was measured using Geriatric Depression Scale and regression analysis was performed. Qualitative part included 21 in-depth interviews analyzed thematically and triangulated with quantitative data. The study revealed that nearly half of empty nesters had depression, with 30% mild and 5% severe. The study identified that single (widowed, divorced or separated) parents had 28.3 times higher cumulative odds of being in a higher category of depressive symptoms, i.e., "moderate/severe and mild" versus "mild and normal" compared to those who are married and living together with their spouses holding constant all other variables. Qualitative interviews supported these findings, highlighting depression among older, widowed parents. The empty nest older population are at increased risk of suffering from depression. Due to the limited evidence, this study investigates if adult children's migration affects the mental health of older parents in Nepal, specifically aiming to understand depression among them and inform policies for their better psychological well-being.

**Data availability statement:** The dataset for the quantitative findings has been included as supplementary file namely, S1 Data. Quantitative data of the study with this manuscript for direct access. An excel file is provided containing coding information for each category of the tool. Also, the qualitative data (codebook) has been included as supplementary file namely, S1 Codebook. Codebook of the qualitative study.

**Funding:** The authors received no specific funding for this work.

**Competing interests:** The authors have declared that no competing interests exist.

## Introduction

Globally, elderly population is increasing rapidly with WHO estimating the elderly population to double between 2015 and 2050 [1]. With increasing life-expectancy, the older population of Nepal has also been increasing rapidly. Nepal census report 2021 reported the older population, i.e., 60 years and above to be 10.21% [2].

Nepal is a South-Asian low- and middle-income country with a high migration level, most of them migrating to seek better employment and education abroad. Absentee population from various census of Nepal shows that many people are separated from their families at different points of life either living within the country or abroad. As per the preliminary report of Nepal census 2021, there are 2.2 million Nepali people living abroad, including 81.28% male and 18.72% female with Kathmandu being one of the top 5 district with highest migration rate. The 2011 census also revealed that one in every four households (25.4%; 1.38 million households) had at least one absent or migrant member [2,3]. Various census data show that the absent population has increased from 1.4 percent of the population in 1942 (absent population question started to be included in the Census) to 7.4 per cent of the population in 2021 [4].

Migration has a tremendous impact not just on the migrants but also on individuals left behind [5]. Children's migration may have both positive and negative impacts on the health and well-being of parents' left behind. Children might boost parents' financial resources and improve access to health and welfare services; however, their absence adversely affect their mental health. Various other studies have also shown that outmigration of adult children has negative effects on parental health outcomes including loneliness, depression and poor psychological health [6–10].

Across South Asia, Nepal in particular, the rise of migration is taking a heavy toll on the mental well-being of older parents. These effects are so deeply rooted in strong cultural norms of filial piety, tradition of living together and proximal caregiving, the physical absence of children often leads to profound loneliness and depression [10,11]. While remittances support them financially and digital communication offer a brief connection, "hands-on" care cannot be replaced- Eg: accompanying them to the hospital. For most parents, the feeling of abandonment overshadows the financial boost ultimately leading to feelings of emotional void left when traditional family roles are disrupted [12].

There is very limited evidence linking adult children's migration to poor mental health outcomes among empty nests older parents who are left behind in Nepal. The older parent's expectations of care deeply rooted in cultural norms of filial piety and in-depth understanding of mental status has not been explored in Nepal. Hence, this study aims to identify if adult children's migration has adverse mental health outcomes like depression, their associated factors among left behind older parents and explore their expectations following migration. This will help in planning effective evidence-based health promoting policies to improve the mental health and psychological well-being of the older parents left behind.

## Objectives

**General objective.** To evaluate the level of depression and the associated factors among older parents' and their expectations after adult children's migration in ward number 9 of Kathmandu Metropolitan City

**Specific objectives.**

1. To assess the depression level among older parents following migration of their adult children.

2. To assess the factors associated with depression among older parents following migration of their adult children.

3. To explore the expectations of empty nests older parents following their children's migration.

## Methodology

### Study design

A cross-sectional study with a concurrent triangulation mixed-method design was used.

### Study site

The study was conducted in ward no. 9 of Kathmandu Metropolitan City. Kathmandu Metropolitan City has the highest household (6.24%) and individual migration rate (6.61%) who are living abroad. Hence, it was chosen for this study [11,13].

### Study duration

The study duration was for 9 months from August 2023 to April 2024. The duration for data collection was for 2.5 months starting from 15th December 2023 to end of February 2024. Approval from ward office of ward no. 9 was obtained for the conduction of the research study on 13th December 2023.

### Sampling procedure and sample size

For the quantitative study, simple random sampling was employed to select the participants. One ward (ward no. 9) was randomly selected from a total of 32 wards of Kathmandu Metropolitan City using the RANDBETWEEN function on MS excel. The researcher contacted the ward chairperson and the ward members and discussed the objectives and implications of the study. After obtaining ethical approval from the IRC, the researcher again communicated with the ward chairperson and staff of ward no. 9 and discussed the methods on how the study will be conducted. The list of above 60 years of age population was used to create a sampling frame of empty nest population. The ward chairperson advised to utilize the list of allowance users to develop a sampling frame as it includes above 60 years aged population as mentioned in the inclusion criteria. All the users of social security allowance schemes (Senior Citizen, Senior Citizen- Dalit, Single Women, Widow, Disability and Malnutrition allowance) were contacted through their phone numbers to identify the empty nests and the sampling frame was developed. The local leaders of Panchakumari Samaj and Mirmire Yuva Club also supported the sampling frame development from the list. Once the sampling frame was developed, simple random sampling (SRS) was utilized and the study samples were selected randomly based on the RANDBETWEEN function on excel within the sampling frame (Table 1).

**Table 1. Sampling Frame for Quantitative Study.**

| S.N. | Ward | Total number of empty nest older population | Number of empty nest older population randomly selected |
|---|---|---|---|
| 1 | 9 | 203 | 140 |

Sample size was calculated by using the following formula:

Sample Size (n) = $z^2pq/d^2$

Here,

Standard Normal Variate (z) = 1.96 at 95% Confidence Interval

Value of p = 0.0818 (Total = 8.18% rate of depression symptoms in empty nesters was 8.18%) [12]

Value of q = 1- 0.0818 = 0.9182

Allowable error (d) = 0.05

Sample size (n1) = 115.4 ~ 116

Again adjusting 10% non-response rate;

Non- response rate = 10% of 116 = 11.6 ~ 12

Sample size (n2) = 116 + 12 = 128

The proposed sample size was 128.

For the qualitative part, judgmental sampling was used as a sampling method for selecting participants for IDI and interviews were conducted till data saturation. Among the participants in quantitative data, 21 empty nest older parents were selected purposively for in-depth interviews. The qualitative component was ensured trustworthy through Guba's four criteria of rigor. Credibility was ensured through rigorous literature-based interview guide, iterative questioning, observation of participant gestures and tone, recording, transcription, translation, and member checking from 3 participants. Dependability was ensured through intercoder reliability (two coders matched 57 codes with an agreement of 85%). Transferability was addressed through detailed contextual and methodological descriptions provided for comparison and replication. Lastly, Confirmability was maintained by bracketing researcher's assumptions, documenting reflexibility documented throughout the study and ensuring that only the data guided the study.

## Study population

For both the qualitative and quantitative study, participants who did not give consent to participate in the study, had severe mental health issues or psychotic conditions, and had communication difficulties due to speaking or hearing impairment were excluded from the study.

**Inclusion criteria.**

1. Participants older than 60 years.

2. Participants who are the permanent residents of the ward.

3. Participants who have at least one born child.

4. Participants who are living alone (with/without spouses) having child(ren) migrated for more than 6 months.

5. Participants whose children migrated internationally.

**Exclusion criteria.**

1. Those who do not consent to participate.

2. Those unable to communicate or those diagnosed with severe psychotic conditions.

3. Those who have at least one child living with the participants.

## Conceptual framework of the study

Fig 1 illustrates the conceptual framework used to analyze the variables in this study.

## Conceptual Framework

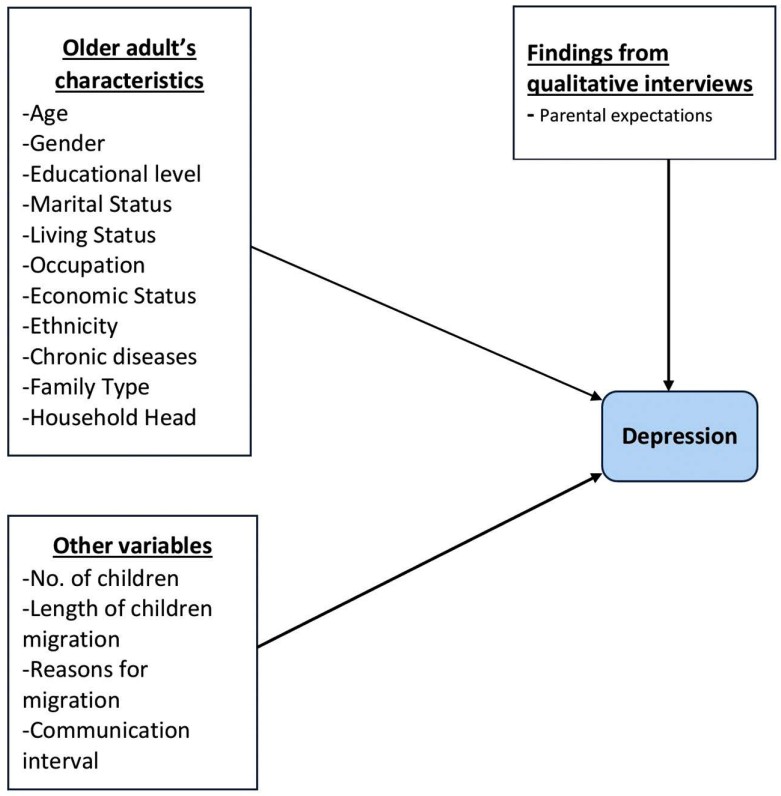

**Fig 1. Conceptual Framework.**

## Data collection and analysis

For the quantitative study, Nepali translated and validated Geriatric Depression Scale (GDS) was used. The data were coded, entered using KoboToolbox, and cleaned in MS Excel 2016. Cleaned data in excel format were imported and analyzed using STATA 13 MP version and EZR software (version 4.0.4). Descriptive statistics were presented depending on the distribution of variables. Mean, median, standard deviation or interquartile range were calculated for continuous variables. Frequency and percentage were calculated for categorical variables. Bivariable analysis of the independent variables was carried out with the dependent variable. Multivariable ordinal logistic regression was carried out for those found significant in bivariable analysis to identify the independent predictor of the outcome variable. Multicollinearity was assessed through Variance Inflation Factor (VIF). In the final multivariable model, only variables with a VIF < 2 and a p-value ≤ 0.25 between the dependent and independent variable in the bivariable analysis were taken to the final model.

For the qualitative study, an in-depth interview guide was developed from an extensive literature review and discussion with the subject expert and the research team. The recorded interviews were transferred to the computer and then first transcribed and translated into the English language. Braun and Clarke's six-step thematic analysis with abductive coding was used to analyze the qualitative interviews [14]. The analysis was based on themes generated. Relevant quotes were used to represent each code within each theme. The codes identified were homogeneous within the theme and heterogeneous between the themes.

**Ethical consideration**

Approval of the study was taken from the Institutional Review Committee of Patan Academy of Health Sciences (IRC-PAHS Ref: PHP2312011820). Approval was also taken from Kathmandu Metropolitan City, Ward No. 9 prior to the conduction of the study. The study's purposes were explained in detail to the participants and informed written consent was obtained from all participants before the interview. Confidentiality of the information was maintained by each participant being assigned separate identification numbers. Likewise, for qualitative data, all the participants were anonymized by using separate respondents' ID as IDI_1, IDI_2, etc. All the data were kept in a password- protected folder and computer with only access to the researcher. The recorded audios will be destroyed after the publication process is complete.

## Results

### Results of quantitative survey

Table 2 outlines the baseline sociodemographic characteristics of the participants. The median age of respondents was observed to be 69.5 years with IQR±7. The majority of them, i.e., 65.71% were 68 years and above which was classified based on the age eligibility criteria of senior citizen allowance in Nepal. Majority of the respondents were females (63.57%) and most of them married. The table also shows that the majority of them were married and living together with their spouses (68.57%). 18.57% of them living with others included house help, caretakers and relatives. Brahmin/Chhetri constituted the most common ethnicity and most of them were living in a nuclear family (84.29%). A total of 68.57% of the respondents were living with their spouses whereas 12.86% of them were living alone and 18.57% of them living with either other relatives or caretakers. Most of them belonged to lower and upper middle socio-economic class calculated based on Kuppuswamy scale. The mean length of migration of the last child was 23 years with standard deviation of 8.32 years classified based on the mean length and most of them having migrated for education purposes. Other reasons for migration included children (daughters) moving abroad after marriage.

Majority of them had some type of chronic disease (77.14%) with hypertension being the most common chronic disease (84.26%). Other chronic diseases included arthritis (5 cases), blindness (1 case), high cholesterol (3 cases), chronic obstructive pulmonary disease (COPD) (15 cases), chronic kidney disease (CKD) (1 case), lung cancer (1 case), partial deafness (1 case), depression (1 case), heart disease (2 cases), ischemic heart disease (2 cases), joint pain (2 cases), migraine (2 cases), nerve pain (2 cases), stroke (2 cases), hemiplegia/paralysis (2 cases), thyroid dysfunction (7 cases).

Item wise responses on the Geriatric depression scale (GDS) showed that the majority of the participants responded to be satisfied with their life (93.57%) with the majority of them not feeling bored and are happy most of the time (88.57%). In addition, 75.71% of them responded to having some sort of memory problems. The total score in each of the responses were added and prevalence of depression was calculated on 4 levels, namely normal, mild, moderate and severe depression. (Table 3)

Almost half of the empty nest respondents had some form of depression with 30% of them with mild depression and 5% of them with a severe form of depression. (Table 4)

The bivariable ordinal logistic regression between sociodemographic variables and depression outcomes showed that sex, ethnicity, marital status, family type, living status, education of the respondents, occupation, household head, socio-economic status, number of children born, number of children abroad, communication interval and presence of chronic diseases were statistically significant with depression.

After assessment of multicollinearity, variables sex, occupation and household head were dropped due to their VIF being greater than 2. Similarly, the variable living status even though very highly correlated (p-value< 0.001) had to be dropped due to high multicollinearity with the marital status variable. Finally, variables with p-value greater than 0.25 and those with VIF less than 2 were taken to the final model (Table 5).

**Table 2. Distribution of participants by Socio-demographic characteristics (N = 140).**

| Variables | Frequency (%) |
| --- | --- |
| **Age** | |
| Md = 69.5 ± 7, Min = 60 years and Max = 84 years | |
| **Age Group** | |
| Below 68 | 48 (34.29) |
| 68 and above | 92 (65.71) |
| **Sex** | |
| Male | 51 (36.43) |
| Female | 89 (63.57) |
| **Ethnicity** | |
| Brahmin/Chhetri | 82 (58.57) |
| Janajati | 57 (40.71) |
| Dalit | 1 (0.71) |
| **Marital Status** | |
| Married/living together | 96 (68.57) |
| Widowed | 42 (30.00) |
| Divorced/separated | 2 (1.43) |
| **Family Type** | |
| Nuclear | 118 (84.29) |
| Joint | 22 (15.71) |
| **Living Status** | |
| With Spouse | 96 (68.57) |
| Alone | 18 (12.86) |
| Others | 26 (18.57) |
| **Education of respondents** | |
| No education | 31 (22.86) |
| Basic education (1–8) | 55 (39.29) |
| Secondary (9–12) | 32 (22.86) |
| More than secondary (13 and above) | 22 (15.71) |
| **Occupation** | |
| Professional/ technical/ managerial | 18 (12.86) |
| Sales and services | 26 (18.57) |
| Skilled manual | 6 (4.29) |
| Unskilled manual | 74 (52.86) |
| Clerical | 1 (0.71) |
| Others | 15 (10.71) |
| **Household head** | |
| Self | 83 (59.29) |
| Spouse | 50 (35.71) |
| Others | 7 (5.00) |
| **Socio-economic status** | |
| Upper Lower | 17 (12.14) |
| Lower Middle | 48 (34.29) |
| Upper Middle | 57 (40.71) |
| Upper | 18 (12.86) |

*(Continued)*

**Table 2.** (Continued)

| Variables | Frequency (%) |
|---|---|
| **Number of children** | |
| Mean = 2.66 ± 0.94 | |
| 1 | 5 (3.57) |
| 2 | 69 (49.29) |
| 3 | 44 (31.43) |
| 4 | 13 (9.29) |
| 5 | 9 (6.43) |
| ≤ 2 | 74 (52.86) |
| 3 and above | 66 (47.14) |
| **Number of children abroad** | |
| Mean = 2.06 ± 0.71 | |
| 1 | 1 (20.00) |
| 2 | 2 (55.71) |
| 3 | 31 (22.14) |
| 4 | 3 (2.14) |
| ≤ 2 | 106 (75.71) |
| 3 and above | 34 (24.29) |
| **Length of last child's migration** | |
| Mean = 23 ± 8.32 | |
| < 23 years | 65 (46.43) |
| 23 years and above | 75 (53.57) |
| **Reason for Migration (Multiple response)** | |
| Education | 135 (96.43) |
| Work | 29 (20.71) |
| Others | 6 (4.29) |
| **Communication Interval** | |
| Everyday | 54 (38.57) |
| Few days a week | 74 (52.86) |
| Once a week | 12 (8.57) |
| **Chronic Disease** | |
| Yes | 108 (77.14) |
| No | 32 (22.86) |

The multivariable ordinal logistic regression revealed that the variable 'marital status' was the only variable that remained significantly associated with depression outcomes after adjustment for other variables. The study revealed that single (widowed, divorced or separated) parents had 28.3 times higher cumulative odds (p-value< 0.001, CI = 9.19-103) of being in a higher category of depressive symptoms, i.e., "moderate/severe and mild" versus "mild and normal" compared to those who are married and living together with their spouses, holding constant all other variables. A proportional odds/parallel test was also conducted in order to check the appropriateness of the ordinal logistic model which showed that the **proportional odds assumption holds** and the effect of marital status is consistent across outcome categories (Table 6).

### Result of qualitative study

The qualitative findings were obtained through thematic analysis of the in-depth interviews. 21 participants were selected purposively for the in-depth interviews. Out of 21 participants, 17 of them were female and the other 4 were male (Table 7).

PLOS Mental Health

**Table 3. Item wise response of Depression based on Geriatric Depression Scale (GDS).**

| S.N. | Items | No | Yes |
|---|---|---|---|
| | | Frequency (%) | Frequency (%) |
| 1 | Are you basically satisfied with your life? | 9 (6.43) | 131 (93.57) |
| 2 | Have you dropped many of your activities and interests? | 92 (65.71) | 48 (34.29) |
| 3 | Do you feel that your life is empty? | 82 (58.57) | 58 (41.43) |
| 4 | Do you often get bored? | 124 (88.57) | 16 (11.43) |
| 5 | Are you in good spirits most of the time? | 16 (11.43) | 124 (88.57) |
| 6 | Are you afraid that something bad is going to happen to you? | 54 (38.57) | 86 (61.43) |
| 7 | Do you feel happy most of the time? | 16 (11.43) | 124 (88.57) |
| 8 | Do you often feel helpless? | 85 (60.71) | 55 (39.29) |
| 9 | Do you prefer to stay at home, rather than going out and doing things? | 126 (90.00) | 14 (10.00) |
| 10 | Do you feel that you have more problems with memory than most? | 34 (24.29) | 106 (75.71) |
| 11 | Do you think it is wonderful to be alive now? | 17 (12.14) | 123 (87.86) |
| 12 | Do you feel worthless the way you are now? | 90 (64.29) | 50 (35.71) |
| 13 | Do you feel full of energy? | 56 (40.00) | 84 (60.00) |
| 14 | Do you feel that your situation is hopeless? | 53 (37.86) | 87 (62.14) |
| 15 | Do you think that most people are better off than you are? | 96 (68.57) | 44 (31.43) |

**Cut off points:**

0–4: Normal/ No depression, **5–8:** Mild depression, **9–11:** Moderate depression, **12–15:** Severe depression.

**Table 4. Prevalence of Depression (N = 140).**

| Variables | Frequency (f) | Percentage (%) |
|---|---|---|
| **Depression** | | |
| Normal | 71 | 71 (50.71) |
| Mild Depression | 42 | 42 (30.00) |
| Moderate Depression | 20 | 20 (14.29) |
| Severe Depression | 7 | 7 (5.00) |

Braun and Clarke's six steps of thematic analysis were followed to identify major themes and codes. The thematic analysis of qualitative interviews identified four themes and their respective sub-themes and codes (Table 8).

**Theme 1: Migration and communication**

**1.1 Place of migration.** Majority of the participant's children had migrated to North American countries like the USA and Canada followed by European countries and Australia.

*All 3 of them are abroad in different places. The youngest is in America, the eldest is in Finland, and the middle one is in Canada. – IDI_17*

*I have two daughters. Both have gone abroad. Both of them are in Australia. One is in Sydney and the other is in Brisbane. – IDI_9*

**1.2 Reason for migration.** The main reason for migration was identified to be for higher education. However, they seemed to settle down abroad after completion of their studies. Very few children went for work purposes and parents with daughters have gone abroad after being married.

**Table 5. Bivariate analysis of socio-demographic characteristics with depression.**

| Variables | Depression | | | Unadjusted OR (95%CI) | p-value | VIF |
|---|---|---|---|---|---|---|
| | f (%) | f (%) | f (%) | | | |
| **Age Group** | | | | | | |
| 68 and above [RC] | 49 (53.26) | 26 (28.26) | 17 (18.48) | REF | 0.448 | |
| Below 68 | 22 (45.83) | 16 (33.33) | 10 (20.83) | 1.29 (0.67-2.49) | | |
| **Sex** | | | | | | |
| Female [RC] | 37 (41.57) | 31 (34.83) | 21 (23.60) | REF | **0.005**\*\* | 3.36 |
| Male | 34 (66.67) | 11 (21.57) | 6 (11.76) | 0.37 (0.18-0.73) | | |
| **Ethnicity** | | | | | | |
| Brahmin/ Chhetri [RC] | 48 (58.54) | 22 (26.83) | 12 (14.63) | REF | **0.022**\* | **1.09** |
| Janajati and others | 23 (39.66) | 20 (34.48) | 15 (25.86) | 2.11 (1.12-4.04) | | |
| **Marital Status** | | | | | | |
| Married/living together [RC] | 70 (72.92) | 22 (22.92) | 4 (4.17) | REF | **0.001**\*\* | **1.95** |
| Widowed and divorced/ separated | 1 (2.27) | 20 (45.45) | 23 (52.27) | 44.1 (16.7-142) | | |
| **Family Type** | | | | | | |
| Joint [RC] | 7 (31.82) | 7 (31.82) | 8 (36.36) | REF | **0.022**\* | **1.14** |
| Nuclear | 64 (54.24) | 35 (29.66) | 19 (16.10) | 0.37 (0.15-0.86) | | |
| **Living Status** | | | | | | |
| With Spouse [RC] | 70 (72.92) | 22 (22.92) | 4 (4.17) | REF | **0.001**\*\* | |
| Alone and Others | 1 (2.27) | 20 (45.45) | 23 (52.27) | 44.1 (16.7-142) | | |
| **Education of respondents** | | | | | | |
| Basic and above [RC] | 63 (57.80) | 27 (24.77) | 19 (17.43) | REF | **0.007**\*\* | **1.27** |
| No education | 8 (25.81) | 15 (48.39) | 8 (25.81) | 2.71 (1.31-5.67) | | |
| **Occupation** | | | | | | |
| Professional/ Sales/ Skilled/ Clerical/ Others [RC] | 41 (62.12) | 17 (25.76) | 8 (12.12) | REF | **0.007**\*\* | 2.30 |
| Unskilled manual | 30 (40.54) | 25 (33.78) | 19 (25.68) | 2.43 (1.28-4.69) | | |
| **Household head** | | | | | | |
| Self [RC] | 33 (39.76) | 28 (33.73) | 22 (26.51) | REF | **0.001**\*\* | 2.77 |
| Spouse and others | 38 (66.67) | 14 (24.56) | 5 (8.77) | 0.32 (0.16-0.62) | | |
| **Socio-economic status** | | | | | | |
| Upper Middle/ Upper [RC] | 50 (66.67) | 18 (24.00) | 7 (9.33) | REF | **0.001**\*\* | **1.32** |
| Upper Lower/ Lower Middle | 21 (32.31) | 24 (24.00) | 20 (30.77) | 4.22 (2.19-8.36) | | |
| **Number of children** | | | | | | |
| ≥ 3 [RC] | 39 (59.09) | 15 (22.73) | 12 (18.18) | REF | **0.122** | **1.55** |
| ≤ 2 | 32 (43.24) | 27 (36.49) | 15 (20.27) | 1.66 (0.88-3.16) | | |
| **Number of children abroad** | | | | | | |
| ≤ 2 [RC] | 48 (45.28) | 36 (33.96) | 22 (20.75) | REF | **0.042**\* | **1.63** |
| ≥ 3 | 23 (67.65) | 6 (17.65) | 5 (14.71) | 0.44 (0.19-0.95) | | |
| **Length of last child's migration** | | | | | | |
| < 23 years [RC] | 32 (49.23) | 22 (33.85) | 11 (16.92) | REF | 0.995 | |
| 23 years and above | 39 (52.00) | 20 (26.67) | 16 (21.33) | 0.99 (0.53-1.87) | | |
| **Reason for Migration (Multiple choice)** | | | | | | |
| Education | 68 (50.37) | 40 (29.63) | 27 (20.00) | | 0.492 | |
| Work | 14 (48.28) | 6 (20.69) | 9 (31.03) | | 0.358 | |
| Others | 4 (66.67) | 2 (33.33) | 0 (0.00) | | 0.318 | |
| **Communication Interval** | | | | | | |
| Everyday [RC] | 20 (37.04) | 18 (33.33) | 16 (29.63) | REF | **0.004**\*\* | **1.33** |
| Few days/ Once a week | 51 (59.30) | 24 (27.91) | 11 (12.79) | 0.39 (0.20-0.74) | | |

*(Continued)*

**Table 5.** (Continued)

| Variables | Depression | | | Unadjusted OR (95%CI) | p-value | VIF |
|---|---|---|---|---|---|---|
| | f (%) | f (%) | f (%) | | | |
| **Chronic Disease** | | | | | | |
| No [RC] | 22 (68.75) | 7 (21.88) | 3 (9.38) | REF | **0.019*** | **1.13** |
| Yes | 49 (45.37) | 35 (32.41) | 24 (22.22) | 2.67 (1.21-6.29) | | |

RC = Reference Category; f= Frequency; OR= Odds Ratio; CI= Confidence Interval; VIF= Variance Inflation Factor.

*p<0.05; **p<0.01.

*Note: All estimates presented in this table are derived from an ordinal (proportional odds) logistic regression model.

**Table 6.** Ordinal (Proportional Odds) Logistic Regression of socio-demographic characteristics with depression (Final Model).

| Variables brought from Bivariable Analysis p<0.25 | Unadjusted OR (95% CI) | p- value | Adjusted OR (95% CI) | p-value |
|---|---|---|---|---|
| **Ethnicity** | | | | |
| Brahmin/ Chhetri [RC] | REF | **0.022*** | REF | 0.712 |
| Janajati and others | 2.11 (1.12-4.04) | | 1.15 (0.53-2.47) | |
| **Marital Status** | | | | |
| Married/ living together [RC] | REF | **0.001**** | REF | **<0.01**** |
| Widowed and divorced/ separated | 44.1 (16.7-142) | | 28.3 (9.19-103) | |
| **Family Type** | | | | |
| Joint[RC] | REF | **0.022*** | REF | 0.182 |
| Nuclear | 0.37 (0.15-0.86) | | 0.50 (0.18-1.39) | |
| Joint | 2.74 (1.16-6.54) | | 2.01 (0.72-5.67) | |
| **Education of respondents** | | | | |
| Basic and above [RC] | REF | **0.007**** | REF | 0.458 |
| No education | 2.71 (1.31-5.67) | | 1.40 (0.57-3.40) | |
| **Socio-economic status** | | | | |
| Upper Middle/ Upper [RC] | REF | **0.001**** | REF | 0.920 |
| Upper Lower/ Lower Middle | 4.22 (2.19-8.36) | | 1.05 (0.42-2.51) | |
| **Number of children** | | | | |
| ≥ 3 [RC] | REF | **0.122** | REF | 0.747 |
| ≤ 2 | 1.66 (0.88-3.16) | | 1.16 (0.48-2.90) | |
| **Number of children abroad** | | | | |
| ≤ 2 [RC] | REF | **0.042*** | REF | 0.635 |
| ≥ 3 | 0.44 (0.19-0.95) | | 0.76 (0.24-2.36) | |
| **Communication Interval** | | | | |
| Everyday [RC] | REF | **0.004**** | REF | 0.064 |
| Few days/ Once a week | 0.39 (0.20-0.74) | | 0.48 (0.22-1.04) | |
| **Chronic Disease** | | | | |
| No [RC] | REF | **0.019*** | REF | 0.176 |
| Yes | 2.67 (1.21-6.29) | | 1.95 (0.76-5.40) | |

RC = Reference Category; OR= Odds Ratio; CI= Confidence Interval.

*p<0.05; **p<0.01.

*Note: All estimates presented in this table are derived from an ordinal (proportional odds) logistic regression model.

**Table 7. Socio-demographic characteristics of the participants.**

| S.N. | Research ID | Age | Sex |
|---|---|---|---|
| 1 | IDI_1 | 67 | Male |
| 2 | IDI_2 | 68 | Female |
| 3 | IDI_3 | 70 | Female |
| 4 | IDI_4 | 74 | Male |
| 5 | IDI_5 | 72 | Male |
| 6 | IDI_6 | 71 | Female |
| 7 | IDI_7 | 60 | Female |
| 8 | IDI_8 | 69 | Female |
| 9 | IDI_9 | 70 | Male |
| 10 | IDI_10 | 69 | Female |
| 11 | IDI_11 | 65 | Female |
| 12 | IDI_12 | 73 | Female |
| 13 | IDI_13 | 69 | Female |
| 14 | IDI_14 | 70 | Female |
| 15 | IDI_15 | 65 | Female |
| 16 | IDI_16 | 70 | Female |
| 17 | IDI_17 | 73 | Male |
| 18 | IDI_18 | 62 | Female |
| 19 | IDI_19 | 70 | Female |
| 20 | IDI_20 | 68 | Female |
| 21 | IDI_21 | 64 | Female |

*They went to study, and after studying, they settled there. Now, after not coming back for so long, they have settled down there. – IDI_20*

The ever-increasing aspirations and desires were another reason for children going abroad rather than going due to poverty or compulsion. Participants also discussed that there has been a trend of moving abroad due to the better earnings and facilities that help them in securing their future. The poor economic situation of the country and unstable government also was identified as a push factor for international migration.

*But now the trend is, one person has gone abroad, another one also went and what would I do here? They see their friend's buying a car, and showing off that's why they also go….., but when you look at the environment, it is top-notch, clean, and you buy a car even if you have to take out a loan for it. So, this is all happening because of human psychology. – IDI_17*

*Now, looking at the current political situation, the situation here is not suitable for living, isn't it? People want to be in a better place, live in a better place, earn a living, and their livelihood is good, so there is nothing particularly negative. -IDI_5*

Another major push factors for migration were lack of employment opportunities in Nepal along with lack of job safety.

*My son went abroad recently. There is nothing here in Nepal, no jobs according to academic qualifications, no income according to jobs. And after getting married, the responsibilities and problems increase, the salary here won't even be enough for pocket money, what to do? So, he had to go, that's all. – IDI_12*

**Table 8. Thematic Analysis Summary Table.**

| S.N. | Themes | Categories (Subthemes) | Codes |
|---|---|---|---|
| 1 | **Migration and communication** | Place of migration | North America, Europe, Australia, Asia |
| | | Reason for migration | Education, Work, Marriage, Rising desires/aspirations, Better facilities/Earnings, Poor economic situation of country, Irresponsible national government, Lack of employment opportunities nationally, Lack of job safety |
| | | Length of migration | Less than 5 years, 5–10 years, 11–15 years, More than 15 years |
| | | Consequences of migration | Only elderly/disabled population reside, Emptying population, Country perishment, Vacant educational institutions |
| | | Communication interval | Everyday, Few days a week, Once a week, Less than once a week, Based on convenience/ leisure |
| | | Means of communication | Internet apps, Cyberbooth phones in the past, Emails/Letters, Ease due to internet |
| | | Frequency of visits | Every 1–2 years, No visits, Self-visits to the place of children, As per their feasibility of both parents and children |
| 2 | **Physical well being** | Physical effects | Accidents, COVID-19, No any |
| | | Chronic Illnesses | Chronic diseases, No any: Old chronic diseases |
| 3 | **Mental well being** | Mental health impacts | Feeling of solitude, Not wish children to go, During natural disaster/Pandemic, Worries/Anxiety, Worries: No phone calls, Stressed: Children's illness, Certain of being unhappy, No any: Devotional activities, No any |
| | | Depression | Perceived absent, Increased when ill, Feeling of empty life, No feeling of empty life, Laziness present, Laziness absent, Good spirit, Afraid of bad happening, Not afraid of bad happening, Happy, Unhappy, Prefer to stay home, Doesn't prefer to stay home, Existence of memory problems, Memory problems: Age related, No memory problems, Worthless feeling, Worthless feeling absent, Energetic, Reduced energy, Hopeless feeling, No hopeless feeling, Feeling others are better off, No feelings of others better off, Past diagnosis, Life satisfaction: Yes, Life satisfaction: No, Activities/Interest: Activities dropped, Activities/Interest: Dropped due to retirement, Activities/Interest: No change from past |
| 4 | **Recommendations** | Advice to empty nests | Seek support from others, Not send children abroad, Self care/ motivation, Being understanding/ considerate, No alternative than to endure, Save money for self, Go and live with children |
| | | Government role | Creating job opportunities, Set pay scale based on qualification, Industrial/Business establishment, Elderly care facilities establishment, Policy changes, Proper implementation of policies, Change/Improve education system, Employment creation through private businesses, Improve health access/sector, Monarchy restoration, No expectations |
| | | Club formation | Necessary, Necessary: Grief sharing, Necessary: Time consumption, Necessary: Self peace/ Happiness/Reduced loneliness, Not necessary: Not feasible in urban area, Elderly bhajan group present |
| | | Elderly allowance | Necessary, Sufficient, Insufficient, Requirement of categorization, Requirement of increment, Need to increase: Cutting down unnecessary allowances, Reduce age eligibility |

*They saw their future abroad. My son says, is a doctor safe and secure in Nepal? How much is the salary? Isn't there any risk here? That is how he convinced me to go abroad. So how can we not let him go? – IDI_11*

**1.3 Consequences of migration.** Various threats of international migration including emptying population and Nepal being the nation where only elderly and disabled population residing were mainly identified. Most of them verbalized that the younger generation are rapidly moving abroad and only the dependent elderly population would reside in the country. The ultimate consequence of the declining youth population being the country being perished and being ruled over by other nations. It was also discussed that the educational institutions are going vacant due to youths migrating internationally serving as a worrying issue nationally.

*Why do so many people have to go abroad for work, even for labor work? Are there any young people left except for the old men and women? Nepal has become an old age home. – IDI_6*

*If this continues, in my opinion, the population will be halved in the next 15–20 years. And then, India will take half and China will take the other half, and the country will be lost. There is a fear that even the name and identity will be lost. – IDI_17*

**1.4  Length of migration.**  The length of migration ranged from less than 5 years to more than 15 years. Most of the participants' children had moved abroad for more than 15 years. The duration of children having migrated very recently was tiny. Most of them had migrated to American and European countries mostly for higher studies and stayed there permanently later.

*My eldest daughter went 23 years ago, my middle daughter went 22 years ago, and my youngest one went 19 years ago. – IDI_6*

*My son went abroad for his education. He's been in America for about 15 years now. – IDI_4*

**1.5  Communication interval.**  Most of the participants had a communication interval of a few days of week followed by many of them talking to their children every day. Only 1 of 22 participants responded that the communication happened less than once a week. While the majority of the respondents communicated a few times a week, few of them also discussed communicating based on the children's convenience and leisure.

*We talk every 2–4 days. There is no time to talk every day. It's not like that abroad, is it? – IDI_12*

*With my son, there might be a gap of around 15 days. I messaged him recently, and he replied saying he's a bit busy, juggling work and studies. When I message him, he says he'll reply when he has free time. And after that, we had a conversation, and everything seems fine till now. – IDI_13*

**1.6  Means of communication.**  The internet applications like Messenger, Viber and Whatsapp were the most commonly used means of communication. These internet applications were perceived to ease the communication contrary to telephone conversations from the cyber booth before the widespread internet access. The internet access was perceived to ease the communication compared to the past.

*We mostly talk through WhatsApp. I usually talk to my daughter through Messenger, and my son through WhatsApp. – IDI_13*

*Nowadays, we live in the era of the internet. Initially, it was quite difficult. We had to go to cyber cafes to make phone calls, which cost a lot of money and time. We were unsure whether we would be able to meet or not. – IDI_9*

**1.7  Frequency of visits.**  Few of the respondents reported their children visiting them every 1–2 years whereas most of them visited based on their feasibility. The parents also verbalized that self-visit was more common compared to children visiting them as children visiting them is quite difficult to manage. Few of them reported that they have been abroad to their children several times which is equally difficult due to the travel.

*They come here once in a while, maybe once a year or once every one and a half years. – IDI_2*

*The children always tell us to come there but it's hard for us to travel. We have already been there 3 times, how many times do we have to go? – IDI_12*

## Theme 2: Physical well-being

**2.1 Physical effects.** Most of them reported to not have any significant physical impacts with only 1 of 22 reporting an accident after their children's migration. Few of the participants reported to have suffered from COVID-19 during the pandemic. Majority of them reported to not have any significant physical effects however had encountered several mental health effects.

*It hasn't affected me physically, but I think it has affected me mentally. – IDI_10*

*Well, there was an accident once, but nothing else. I slipped and hurt my leg, but I didn't tell my son about it. (laughs) They worry too much, my husband was also there at that time and I recovered by taking care of myself at home. – IDI_3*

**2.2 Chronic illnesses.** Most of them reported to have suffered from chronic diseases like diabetes and hypertension due to continuous stress and worries following migration of their children. However, among those with chronic diseases, the majority of them informed about the pre-existence of chronic diseases before their children's migration. Most of them had illnesses when their children were back in Nepal with them.

*Now I'm worried because he's alone abroad, and I've even developed diabetes. I'm also taking medicine for diabetes because of the stress. – IDI_7*

## Theme 3: Mental Well-Being

**3.1 General mental health impacts.** In general, most parents did not express any mental health impacts due to their children's migration. However, many of them verbalized that they did not want their children to go abroad at the first place with most of them expressing worries after their children's migration due to them having migrated at a very young age. Most parents also revealed that their worries only escalated when their children fell ill.

*I don't have any effects. My children take care of me, they ask me what I have eaten, what I have taken, and where I have gone. And my sons keep me happy. – IDI_19*

*Mentally, I haven't been affected that much since my children left. Sometimes when I am with my family, I sometimes wish she had not gone. When we are sick or doing something big, all parents miss their children and I do too. – IDI_18*

*I am a mother so, when he tells me he is ill, I do get a little worried and panicked sometimes. – IDI_7*

One very crucial information revealed in the study was that no parents would ever be happy due to their children moving abroad. Few parents also concluded that children's migration has severe mental health impacts and the truth being all the parents feeling miserable even if they don't seem to verbalize it blatantly.

*But in reality, in the end, it is true that no parents are happy after sending their children abroad. This is the ultimate truth. That's the conclusion you have to come to. I can bet on it. – IDI_11*

*It's not a good thing for children to go abroad. We live here and talk about lakhs and crores, my son is in America, UK etc. but just money doesn't give satisfaction. There is nothing bigger than being able to live with your family. I feel bad about that. – IDI_18*

Few of the elderly parents perceived to feel abandoned while the majority did not have such feelings. However, 8 of them verbalized that they are quite certain of loneliness feeling to increase in future when they are much older and are subjected to severe health ailments.

*Now that we are old and we have no one to cook food for us, even if we have money, we don't have people to work in the fields in the village. Now, even in my share, I have 55–56 ropani of land. What do we do about it, it has been left barren. There are no people to work in the fields. – IDI_4*

*It was a little lonely for some time, I felt bad. Because so far, our health is good, so it's okay. Tomorrow, if our health deteriorates and the two children are abroad, it will be difficult in old age. That's true. Who is going to look after us, who is going to take us to the hospital, but so far, it's okay. – IDI_9*

**3.2 Depression.** Most parents revealed that they felt sad primarily when they themselves fell sick or have to go to the hospital. 8 out of 21 parents felt a sense of emptiness while the other did not feel any sense of emptiness due to their children's migration. Majority of the parents reported that they did not experience any laziness or wanting to stay back home. Most of them mentioned that they do all the household work enthusiastically with few of them also working outside homes. Only one of them expressed to have experienced laziness and lack of will to go outside the house.

*Mentally, I haven't been affected that much since my children left. Sometimes when I am with my family, I sometimes wish she had not gone. When we are sick or doing something big, all parents miss their children and I do too. – IDI_18*

*I don't really feel lazy. I mean, it's normal for people to feel lazy sometimes, but I don't feel that lazy. I still cook my own food, go to the neighborhood circle, and do everything. I am content with everything. – IDI_19*

While most of them stated to be happy, the constant fear of bad events happening was equally present among the empty nest parents. Many of them confessed to the presence of constant fear and stress of bad events like accidents and illness occurring while about half of them did not feel any sort of fear about occurrences of such mishaps.

*Yes, it did. After he went abroad, I felt more tense. Don't you hear all sorts of things about being abroad? I heard that someone died while swimming. Even though it is not my own son but I panic, especially when he is sick. Even when he had COVID twice, I was thinking if he was close to me, I would have taken more care of his health and provide him with healthcare. – IDI_7*

*That fear has always been there. It has always been there. I worry that he might get sick or get into an accident. Even if something happens, it is not easy to go there. That fear was there when he went abroad and it is still there now. – IDI_10*

Memory problems were quite existent among the parents with the majority of them reporting to have forgetfulness to a considerable degree. However, most of them perceived the issue to have occurred due to old age rather than as a consequence of stress due to their children's migration.

*That's a bit less. Forgetting a bit is natural, isn't it? Also, with age, there's more now than before. Remembering certain things, like the names of old friends, might require more effort; forgetting happens, that's natural. – IDI_5*

Majority of the parents also revealed that they do not have any worthless feelings towards themselves or that others are more capable than themselves. Majority of them stated to be satisfied with their life in general. Only 3 of them confessed

to feeling worthless and could have done better in life compared to others. Most of the participants reported to have life satisfaction with only one of them reporting to not be satisfied in life.

*I feel that very much, my dear (laughs). I feel that every time. Alas, I did not use my brain on time, if I had studied a little bit, I would have been fine. It's not that my parents didn't teach me, but I just didn't care. – IDI_13*

*Oh, I am very content. I am very satisfied. I haven't faced any significant problems in life, neither from any life event nor from my children. – IDI_2*

Majority of them reported to still be quite energetic and actively engage in various activities. 6 of the respondents however reported to have reduced energy mostly due to chronic illnesses occurring at old age. Most of the participants also reported to continue doing the activities like in the past with no any changes while few of them have stopped doing past activities mainly due to retirement.

*I have to say that I am still active. My body is quite huge to look at. I might not look but I am still very active and energetic. – IDI_16*

*Well, now that I'm older, my eyesight has weakened, even walking is improper and I am unable to work like before. – IDI_4*

**Theme 4: Recommendations**

**4.1  Advice to empty nests.**  Most of the parents advised the empty nests to be understanding of the situation, understand their children and be considerate as there is no other option than to endure and live through the situation. Self-motivation was widely advised along with being self-sufficient to not expect anything from others, including their own children. One of them also advised me to save up some money for myself for the older life ahead. Lastly, if the situation is relatively tough, one of them is also advised to go abroad and live with children.

*The advice that I would like to give them is, understanding that the happiness of sons and daughters is their own happiness, their happiness is our happiness, that's all I want to say. – IDI_2*

*The advice I would give to them is not to expect too much from their children till you can work yourself. As much as you can, try to be self-sufficient. Until the time you can take care of yourself, don't expect too much from your children. – IDI_8*

**4.2  Government role.**  Creating job opportunities also with establishment of private companies and setting a standard pay scale based on qualification were mostly identified to be a major role of the government. One of the major reasons for international migration was identified to be due to lack of proper employment opportunities. Many parents agreed that the national policies, even though they are well written, are not well implemented. Hence, there must be proper implementation of policies along with improvement of education and health sector. Industries and local businesses establishment were necessary which also helps in employment creation. This was perceived to help in retention of the youth in the country and minimizing migration. Establishment of elderly care facilities with improvement in senior citizen facilities were also emphasized in the interviews. One of the parents also mentioned monarchy restoration to be necessary contemporarily.

*If the government had provided good employment, this situation wouldn't have happened. Now, only old people like us are left living in Nepal. They would have stayed here if there was good employment. – IDI_16*

*The government should open up more opportunities, right? They should open up more industries and businesses, and do development work. They should create an environment where educated youth can be engaged. – IDI_10*

**4.3 Club formation.** Elderly gathering club was deemed to be necessary by the majority of the parents as being involved in such clubs helps in depressive symptoms reduction through grief sharing with other empty nest parents. It was also perceived that parents gathering in such club is vital in forgetting their worries and also as a time pass. However, one of them stated that formation of such clubs would not be feasible and necessary in an urban area like Kathmandu. The study area also consisted of a bhajan group for elderly which functioned every day in the evening after 4 pm. This was reported to help all the elderly and many of them visited the group every day.

*It is very necessary. Because if many elderly people come together and open such a club, they will have friends to share their sorrows and joys with, they will be able to talk and gossip with friends, and they will not feel lonely. – IDI_15*

*Well, I don't feel the need for such things in Kathmandu right now. There doesn't seem to be a need. In the past, everyone used to gather together in the village, like elderly folks sitting together in the courtyard and chatting. That was possible. But in today's situation, it's not possible. Now, even meeting neighbors in the neighborhood seems difficult. Joining a club or something similar seems impossible. – IDI_1*

**4.4 Elderly allowance.** The senior citizen allowance of Rs. 4000 was deemed to be extremely necessary for the elderly as most of them do not have an income source and it aids in purchasing essential resources like food and medicines. Majority of them however reported that the amount is not sufficient at the current situation and must be increased by any means including cutting down unnecessary allowances of the political figures. There must also be an accurate categorization of elderly who are poor and actually are in need of the allowance. Most of them confessed that they do not actually require the allowance but the government does not have criteria for identifying those in need. Also, few of them also stated that the age eligibility criterion of 68 years is too far-fetched as most women succumb to chronic diseases after menopause and hence the age eligibility criteria should be reduced.

*They are not sufficient, not enough. Because I have to take medicine, and the 4000 allowance is only enough for me to buy my medicines. That allowance is not enough. Many treatments should be made free, only those who can afford can pay. – IDI_15*

*Are you going to sit there and wait for your money to come at 70 because you can't afford to buy medicine when you get sick? That is not okay. We have to make women strong until 70. That's why I think whatever I say, that social security allowance should be given to women at the age of 55, most women get sick at the age of 55. – IDI_11*

**Results of data triangulation.** Data triangulation is the practice of using multiple sources of data or multiple approaches to analyze data to increase the credibility of the research study. Here, we have used the data from the quantitative and qualitative study to identify and validate the factors associated with depression among the empty nest older population.

## Convergent findings

a. **Marital Status:** The quantitative study identified that for the widowed, divorced or separated empty nest parents the odds of being likely (i.e., moderate/severe and mild vs normal) to suffer from depression was 28.3 times (p-value< 0.001, CI = 9.19-103) compared to those who are married and living together with their spouses. This finding was similar to that of the qualitative study which also reported increased depressive symptoms due to spouse's death.

b. **Reason for Migration:** The quantitative study showed that the majority of the participant's children moved abroad for educational purposes (96.43%). This finding is similar to the majority of responses in the qualitative interviews which also revealed the major reason for migration to be for education.

*They went to study, and after studying, they settled there. Now, after not coming back for so long, they have settled down there. – IDI_20*

c. **Length of Migration:** The quantitative study showed that the mean length of migration was 23 years with more than half (53.57%) having migrated for more than 23 years. This was also reported in the qualitative study where most of the participants' children had migrated for more than 15 years.

d. **Communication Interval:** The quantitative study showed that most of them (52.86%) communicated with their children every few days a week. This is also corroborated by the qualitative interviews which revealed most of the parents communicating with their children every few days a week.

*We talk every 2–4 days. There is no time to talk every day. It's not like that abroad, is it? – IDI_12*

e. **Life Satisfaction:** It was identified that the majority of the participants responded to be satisfied with their life (93.57%). This finding was also reported in the qualitative interviews where most parents verbalized to be satisfied with their life.

*Oh, I am very content. I am very satisfied. I haven't faced any significant problems in life, neither from any life event nor from my children. – IDI_2*

f. **Activities Dropped:** It was identified that most of the participants responded to not having dropped their usual activities following migration of their children (65.71%). This finding was also reported in the qualitative interviews where most parents verbalized to have continued doing the past activities.

*No, there's nothing like that. I still do whatever I want to do. It's not like I stopped doing things just because my son went abroad. – IDI_7*

g. **Feeling Bored:** It was identified that the majority of the participants responded not feeling bored (88.57%). This finding was also reported in the qualitative interviews where most parents reported to not feel lazy and do their regular activities efficiently.

*Lazy? Not really. You cannot afford to be lazy doing the household chores. Well due to age sometimes I don't feel like doing work but I don't like to just sit around and be lazy. If I have time to go out and do something, I do that. I don't want to just sit at home and do nothing. I have been able to do it till date. – IDI_13*

h. **Fear of Bad Happening:** The quantitative study showed that most of them (61.43%) feared of something bad happening. This is also corroborated by the qualitative interviews which revealed most of the parents having a bad feeling of adverse events to occur anytime.

*That fear has always been there. It has always been there. I worry that he might get sick or get into an accident. Even if something happens, it is not easy to go there. That fear was there when he went abroad and it is still there now. – IDI_10*

i. **Prefer to Stay Home:** It was identified that the majority of the participants responded not feeling lazy and stay home (90%). This finding was also reported in the qualitative interviews where most parents reported they would rather go out often than staying home alone.

*I don't really feel lazy. I mean, it's normal for people to feel lazy sometimes, but I don't feel that lazy. I still cook my own food, go to the neighborhood circle, and do everything. I am content with everything. – IDI_19*

j. **Memory Problems:** The quantitative study showed that the majority of them (75.71%) of them had some sort of memory problems like forgetting. This is also corroborated by the qualitative interviews which revealed most of the parents often being forgetful mostly due to age factor.

*That's a bit less. Forgetting a bit is natural, isn't it? Also, with age, there's more now than before. Remembering certain things, like the names of old friends, might require more effort; forgetting happens, that's natural. – IDI_5*

k. **Energetic:** The quantitative study showed that most of them (60%) of them still felt that they were energetic. This is also revealed in the qualitative interviews where the parents shared that they are still very active and are full of energy.

*I have to say that I am still active. My body is quite huge to look. I might not look but I am still very active and energetic. – IDI_16*

### Divergent findings

Family type was identified to be the area of divergence in the study. Similarly, while assessing depression responses on each item of the GDS scale, there was no area of divergence.

a. **Family Type:** The quantitative study identified that for those living in a joint family, the odds of being likely (i.e., moderate/severe and mild vs normal) to suffer from depression was 2.74 times (CI: 1.16-6.54) compared to those living in a nuclear family. However, the qualitative study revealed that those living in extended or joint families were less likely to get depressed as they are always around people and have someone to talk to.

*I didn't feel lonely because I have a joint family and I have a lot of responsibilities. I love him and missed him, but I didn't feel alone. I never felt that way. The atmosphere in my house is very different, so I didn't have a chance to feel that way. I was never alone, not alone even in a flat. (laughs) So that's why. – IDI_10*

### Expansive findings

a. **Place of Migration:** North American countries like USA and Canada followed by European countries and Australia were identified to be the most common place of migration.

b. **Consequences of Migration:** Consequences of migration are threatening including emptying population and Nepal being the nation where only elderly and disabled population reside ultimately leading to the country being perished.

*Now there is a little hope that the new ones will come and do something. Now we have to think about this issue otherwise there will be no one left in Nepal except the disabled and elderly men and women in the next 10–12 years. There will be no one except the helpless old men and women. – IDI_18*

c. **Means of Communication:** The internet applications like Messenger, Viber and Whatsapp were the most commonly used means of communication. These internet applications were perceived to ease the communication compared to the past.

*We mostly talk through WhatsApp. I usually talk to my daughter through Messenger, and my son through WhatsApp. – IDI_13*

d. **Financial Challenges:** No financial hardships were faced by the parents following their children's migration with very few of them expressing the challenge.

*Nothing happened. The children went on scholarship, they didn't ask me money at all. Later when my daughter had to pay for college, she didn't even ask me to give her money. – IDI_13*

e. **Emotional Challenges:** The study revealed that few of the parents expressed to face emotional challenges following their children's migration aggravated by their spouses' death. Majority of them however did not express the presence of any emotional challenges of significant level.

*Main challenge was when their father passed away. – IDI_3*
There were none. Whether it is physical or financial, there are no challenges. I didn't have any financial problems. – IDI_17

f. **Advice to Empty Nests:** The study also explored what the empty nest parents would suggest to other empty nests. It was identified that enduring the situation, self-motivation and being understanding are necessary.

*If you have children who listen, you don't need to give them advice. You have to be strong on your own feet and you need to have some money with you. If you don't have that, nothing will work, and if you do, no one can do anything to you. – IDI_16*

g. **Government Role:** The study recommended major government roles to be job creation, factories/industries establishment, proper implementation of policies, increase in elderly care facilities, etc.

*Well, the government's policy is good. There is a policy to create employment, even a policy to open factories in Nepal in the future, but there also needs to be good governance in that. – IDI_1*

h. **Elderly Allowance:** The study identified that the elderly allowance was extremely necessary however, insufficient. It was recommended to be increased with decrease in age eligibility criteria.

*They are not sufficient, not enough. Because I have to take medicine, and the 4000 allowance is only enough for me to buy my medicines. That allowance is not enough. Many treatments should be made free, only those who can afford can pay. – IDI_15*
*Are you going to sit there and wait for your money to come at 70 because you can't afford to buy medicine when you get sick? That is not okay. We have to make women strong until 70. That's why I think whatever I say, that social security allowance should be given to women at the age of 55, most women get sick at the age of 55. – IDI_11*

i. **Elderly Club Formation:** Elderly gathering club was deemed to be necessary as such clubs helps in depressive symptoms reduction through grief sharing with other empty nest parents. It was also perceived that parents gathering in such clubs is vital in forgetting their worries and also as a time pass.

*It is very necessary. Because if many elderly people come together and open such a club, they will have friends to share their sorrows and joys with, they will be able to talk and gossip with friends, and they will not feel lonely. – IDI_15*

j. **Depressive symptoms increased when ill:** The study revealed that most parents felt sad primarily when they themselves fell sick or had to go to the hospital.

*Sometimes when I am with my family, I sometimes wish she had not gone. When we are sick or doing something big, all parents miss their children and I do too. – IDI_18*

**Joint display for the major convergent findings:**

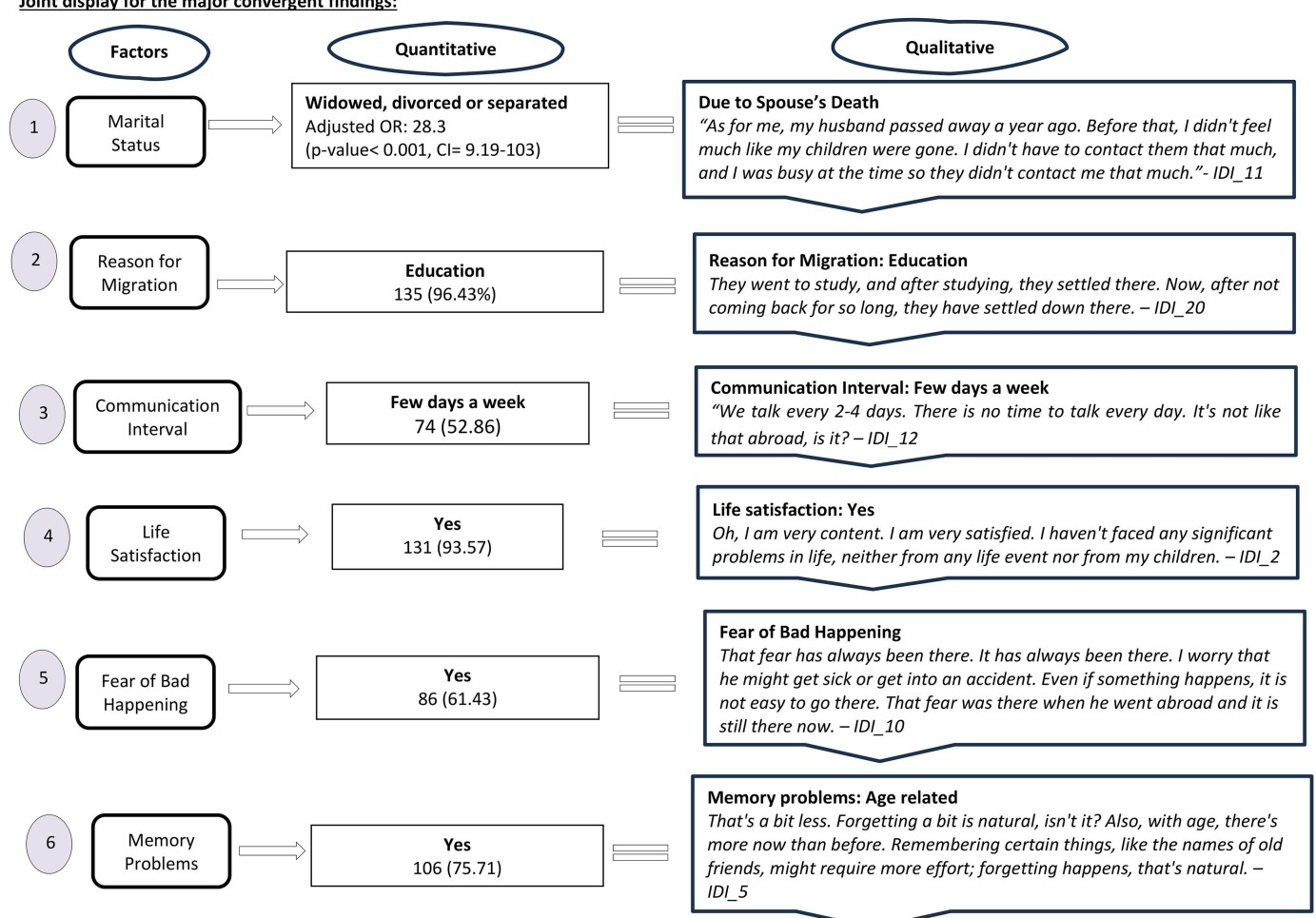

## Discussion

According to this study, the prevalence of depression among empty nest elderly parents was 49.29% with 30% of them suffering from mild depression, 14.29% of them with moderate form of depression and 5% of them having severe form of depression. The prevalence of depression was almost 11 times higher when compared to the findings of National Mental Health Survey 2020 which identified the lifetime prevalence of major depressive disorder to be 4.6% among the above 60 years of age population. This study corresponds with the findings of NMHS 2020 where depression was more prevalent among females compared to males [15].

Another study conducted among geriatric population in Nepal showed the prevalence of depression to be 53.2% with 34.2% of them suffering from mild form of depression and 19% with severe form of depression. This finding is similar to our study which found the prevalence of depression to be 49.29% [16]. Another study among elderly parents found the prevalence of depression to be 36.1% among the elderly parents of migrant households [11]. Likewise, the prevalence was also found to be similar with a systematic review of mental disorders among older people in Nepal which reported the prevalence of depressive symptoms to range from 47.3% to 72.8% [17].

Left-behind older parents are identified to be at higher risk of depression as evidenced by other studies cross South-East Asia [18,19–22]. However, another perspective emerges in other studies which report that adult children's migration

may not pose any psychological impacts including depression. A cross-sectional study conducted in a community in Nepal showed that adult children's migration was not associated with inverse health outcomes including depression among the left-behind elderly parents [15].

Overall, the findings of our study suggest that there is a high prevalence of depression among the empty nest elderly population which is not included in any national level surveys in Nepal. National level mental health survey reports higher prevalence of depression among elderly population with few studies reporting similar findings among the empty nest population in Nepal [15].

While assessing factors associated with depression, it was found to be significantly associated with the marital status of the older population in this study after controlling for all the other covariates. The widowed/divorced and separated empty nests were 28.3 times more likely to suffer from depression (i.e., moderate/severe and mild vs normal) compared to the married ones living with their spouses. The qualitative study reported convergent findings where those widowed reported higher feelings of depression. Another study in Nepal also reported a similar finding where marital status was found to be statistically significant with depression among the empty nest elderly population. However, only bivariable analysis was conducted in this study without controlling other covariates [9]. Various other studies suggest that marital status is a predictor of depressive outcomes. A meta-analysis among empty-nesters in China revealed that widowed empty nest parents are 1.75 times (CI: 1.29-3.28) likely to be depressed which is consistent but much lower with the finding of our study [21]. A cross-sectional study in China also reported that the widowed or divorced empty nests are 1.27 times (CI: 0.67 to 2.39) likely to get depressed than the married ones.[17] Many other studies report similar findings, where being currently married is associated with better mental health outcomes including depression among the left-behind older parents [22–24]. A meta-analysis study revealed that depression was more likely among those who are living alone [25].

This could also be because married empty nesters have spouses living with them and rarely get depressive symptoms [26]. This could be because spouses provide emotional support during tough times ultimately reducing depressive symptoms. In contrast, absence of a spouse leads to feelings of emptiness exacerbating the depressive symptoms among the elderly. Marriage provides vital emotional, practical, and social support, often buffering stress and fostering resilience. When a spouse is absent—especially in empty-nest households where children live elsewhere—older adults face a "double loss," increasing loneliness, role strain, and psychological vulnerability. In collectivist societies, co-residence with a spouse and children is culturally valued, so spousal loss can further weaken social integration, perceived status, and daily support. At the same time, social participation and community engagement can help mitigate these effects. These findings underscore the central importance of relational support for mental health in older adults, illustrating how cultural, social, and familial contexts shape vulnerability to depression and highlighting avenues for supportive interventions [27–30].

While digital communication offers comfort, perceived emotional support predicts psychological well-being better than mere contact frequency [10,12]. Moreover, virtual interactions cannot compensate for the "care drain" and loss of instrumental physical assistance caused by migration suggesting communication interval does not suggesting communication interval does not adequately offset the profound emotional and physical void left by migrating children [31,32]. Aligning with these literatures, our final model also revealed that communication frequency does not independently predict depression among left-behind parents.

## Strengths and limitations of the study

The major strength of the study is the use of correct statistical technique (ordinal logistic regression) making the results of the study valid. The mixed method approach provided a more comprehensive and nuanced understanding of factors associated with depression among the empty nest elderly population by combining the strengths of both qualitative and quantitative methods. It further provides a more complete picture of the phenomenon being studied, by capturing both the subjective experiences and perspectives of the participants (qualitative) and the prevalence and associations of the phenomenon (quantitative). The use of simple random sampling for selecting the ward and the participants increased the

representativeness of the sample. However, a sampling frame was developed based on social security allowance lists and household listing was not done which might pose some selection bias.

Moreover, the use of intercoder percentage agreement and participant checking helped enhance the validity and reliability of the findings, by reducing the risk of researcher bias and ensuring that the data are accurately and consistently coded and interpreted. However, the subjectivity may not have been avoided completely during qualitative data analysis and interpretation. Also, the study was not guided by a culture-grounded theoretical framework at data collection, limiting in-depth analysis of underlying mechanisms. As the study relies on self-reported data, there may have been issues with recall bias, social desirability bias, and underreporting of sensitive information. In regards to specific variables such as family monthly income and length of migration where the participants may either not provide true information of the income and cannot recall the exact years of migration.

This was a cross-sectional study due to which causal inferences cannot be made. Hence, longitudinal studies should be conducted to establish causal inferences between socio-demographic factors with depression. Because the sampling frame was based on the social security allowance registry, which provides benefits from age 60 only to selected groups (e.g., Dalits, marginalized ethnic groups, and widows) and from age 70 (previously 68) to others, adults aged 60–69 from non-eligible groups were underrepresented, limiting the representativeness and generalizability of our findings to all older adults in Nepal. Also, the lack of data on social support network accessibility may restrict our ability to completely comprehend their possible impact on depression outcomes among parents who are empty-nesters. Additionally, while material support (e.g., remittances) is a recognized factor in migration studies, it was not specifically queried during the study. Therefore, the contextual nuances of how financial support- or the lack thereof- affects the psychological well-being of left-behind parents remain an area for future exploration. Lastly, this study was conducted in affluent areas of Kathmandu Metropolitan City. Hence, the findings may significantly differ when conducted among impoverished communities.

## Conclusion

Depressive symptoms were found to be quite prevalent among the empty nest population with almost half of the participants having some form of depression. The present study found that widowed, divorced and separated empty nest parents who are living alone are at increased risk of developing depression. This finding was also corroborated by the qualitative study conducted concurrently which revealed that death of a spouse was a major reason for increase in depressive symptoms.

## Policy recommendations

The conclusions drawn from this evaluation carry significant significance for initiatives and regulations designed to promote the psychological well-being of senior citizens. Focusing on promoting social security measures for the elderly who may feel neglected could foster a sense of reassurance and assistance, consequently enhancing their mental health and overall welfare.

Based on the findings of the study, it is strongly recommended that it is absolutely necessary to identify the root cause of international migration. There are numerous government's roles that need to take place in order to reduce the migration rate. The push factors should be reduced by an increase in pull factors. First pull factor would be creation of job opportunities especially based on qualification with standard pay scale. This can be achieved through establishment of industries, factories and private businesses. Several guiding laws and policies exist in Nepal regarding migration in Nepal. The constitution of Nepal, Foreign Employment Act 1985, Foreign Employment Policy 2012 along with Immigration Act 2049 (1992) exist in Nepal governing laws regarding movement and safe working environment inside and outside the country. However, the study recommends the proper implementation of such policies.

There are few notable governmental programs targeted to the older population in Nepal including senior citizen allowance, free healthcare and pensions. However, not everyone benefits from such schemes as most health care institutions

in Nepal are privately owned. There are no government owned old aged homes specifically for the older population who are living alone. The government should effectively implement the allowance program to every population in need along with expansion in health care services especially catered towards them. Another finding includes the elderly allowance to be inadequate considering the economic inflation. Hence, an increase in the allowance amount is recommended especially to those who are also economically vulnerable. If possible, the government should methodologically categorize those in need of the allowance. The findings of the study suggests that it is absolutely necessary to identify mechanism and supportive interventions for community and societal collaborative programs focused on getting the empty nests parents come together to alleviate the absence of proximal care and bring about targeted interventions to minimize the mental toll of loneliness due to children's absence.

## Supporting information

**S1 Data. Quantitative data of the study.**
(XLSX)

**S1 Codebook. Codebook of the qualitative study.**
(XLSX)

## Acknowledgments

We would like to express our sincere thanks to the Institutional Review Committee of Patan Academy of Health Sciences for providing the ethical approval to conduct this study. Finally, we would like to thank the ward office of Ward Number 9, Kathmandu Metropolitan City for their support in sampling frame development and participant recruitment.

## Author contributions

**Conceptualization:** Soniya Shrestha.

**Data curation:** Soniya Shrestha.

**Formal analysis:** Soniya Shrestha.

**Investigation:** Soniya Shrestha.

**Methodology:** Soniya Shrestha.

**Software:** Soniya Shrestha.

**Supervision:** Soniya Shrestha, Sapana Ghimire, Madhusudan Subedi.

**Validation:** Soniya Shrestha.

**Visualization:** Soniya Shrestha.

**Writing – original draft:** Soniya Shrestha.

**Writing – review & editing:** Soniya Shrestha, Bigya Tuladhar, Yamuna Chhetri, Sapana Ghimire, Madhusudan Subedi.

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
