## [Decision Letter · Decision Letter 0]

11 Mar 2025

PMEN-D-24-00540

Depression among older parents’ and their associated factors after adult children’s migration: A mixed method study in an urban setting of Kathmandu Metropolitan City

PLOS Mental Health

Dear Dr. Shrestha,

Thank you for submitting your manuscript to PLOS Mental Health. I apologise for the delays. After careful consideration of the reviewer reports, we feel that it has merit but does not fully meet PLOS Mental Health’s publication criteria as it currently stands. Therefore, we invite you to submit a revised version of the manuscript that addresses the points raised during the review process. Please address all comments raised by the reviewers, which you can find below.

We look forward to receiving your revised manuscript.

Kind regards,

Karli Montague-Cardoso

Executive Editor

PLOS Mental Health

Journal Requirements:

1. In the online submission form, you indicated that “The data will be made available upon request to the corresponding author.”.

a. In a public repository,

b. Within the manuscript itself, or

c. Uploaded as supplementary information.

Additional Editor Comments (if provided):

Reviewers' comments:

Reviewer's Responses to Questions

**Comments to the Author**

1. Does this manuscript meet PLOS Mental Health’s publication criteria? Is the manuscript technically sound, and do the data support the conclusions? The manuscript must describe methodologically and ethically rigorous research with conclusions that are appropriately drawn based on the data presented.

Reviewer #1: Yes

Reviewer #2: Partly

2. Has the statistical analysis been performed appropriately and rigorously?

Reviewer #1: Yes

Reviewer #2: No

3. Have the authors made all data underlying the findings in their manuscript fully available (please refer to the Data Availability Statement at the start of the manuscript PDF file)?

Reviewer #1: No

Reviewer #2: Yes

4. Is the manuscript presented in an intelligible fashion and written in standard English?

Reviewer #1: Yes

Reviewer #2: No

5. Review Comments to the Author

Reviewer #1: The manuscript addresses an important mental health issue among older adults, particularly in the context of empty nest period. It employs a mixed-method approach to examine depression among empty-nest elderly parents. With a good revision, I agree that the paper can contribute well to the literature. I have several comments to improve the manuscript further:

1. The manuscript lacks a strong theoretical framework to guide the study. While it discusses the negative effects of empty nest on elderly parents, it does not integrate established psychological mechanisms to explain why and how empty nest might lead to depression. Also, the study appears to assume that adult children's migration is inherently distressing for parents, but it does not sufficiently account for potential positive psychological adaptation. Some parents might experience role relief and increased autonomy despite physical separation. Cultural contexts may play important role in this discrepancy. I would strongly suggest the authors to expand their literature review, provide a more balanced review of the literature, and improve the theoretical foundation. This would strengthen the introduction of the paper. I found a recent theoretical paper on empty nest that is relevant and can help to improve the introduction: Cultural contexts differentially shape parents’ loneliness and wellbeing during the empty nest period. Communications Psychology, 2(1), 105.

2. The study should also address whether pre-existing health conditions, socio-economic status, or access to social support networks influenced the depression outcomes. If not possible to be included in the analysis, this should be highlighted as limitation.

3. Regarding the use of GDS, the cutoff points for mild, moderate, and severe depression should be explicitly stated. Does the study use an internationally accepted threshold?

4. For the analysis, the authors should provide further justification of why certain independent variables excluded from the final model despite their statistical significance in bivariate analysis?

5. The qualitative findings are rich but should be better integrated with the quantitative data. How do the lived experiences of parents (from in-depth interviews) align or contradict the statistical findings?

6. For the thematic analysis, there is no discussion on intercoder reliability. There should be more justification on how was bias minimized in qualitative coding?

7. The abstract should include the quantitative and qualitative sample sizes

8. The discussion is overly descriptive. It should focus more on mechanisms of depression rather than simply repeating the results.

Reviewer #2: Thank you for providing the opportunity to review the manuscript “Depression among older parents’ and their associated factors after adult children’s migration: A mixed method study in an urban setting of Kathmandu Metropolitan City”. This manuscript explores an important and timely issue—the impact of adult children's migration on the mental health of left-behind older parents. While the study addresses a relevant topic, several aspects of the manuscript require substantial revisions for clarity, coherence, and methodological rigor. Key issues include unclear definitions, inconsistencies in the objectives, methodological ambiguities, and insufficient engagement with existing literature. Additionally, the results and discussion sections need refinement to ensure alignment with the study’s stated aims. Specific concerns regarding statistical analyses, conceptual framework, and qualitative findings also need to be addressed.

Abstract:

What does 2.2 absentee population mean in the background section of the abstract? Is this 2.2% or 2.2 million? Please clarify. Furthermore, I would like to see something on mental health and depression among the older people in the background to justify the study.

The results section should be re-written. Maybe it is better to write single (widowed, divorced or separated) vs. married. Also, what is the outcome variable not clear.

How did the qualitative findings support these findings?

Please consider re-writing the conclusion too. “The empty nest older population are at higher risk of suffering from depression.” Compared to what? Did this study compare empty nesters vs. non-empty nesters? There are no such findings reported in the results section. To conclude “……. Adult children’s migration affects the mental health of older parents…………”, you should have comparison group. Additionally, what are other mental health conditions?

Introduction:

What is the definition of older population or elderly in Nepal?

If 7.4% of the population in Nepal are absent, how could a district (Kathmandu) with 6.56% absent population be the district with highest migration?

Please clarify whether absentee and migrants are same.

Line 72-73: How can financial resources and improved access to health and welfare services adversely affect their mental health?

Line 76-77 “There is very limited evidence linking adult children’s migration to poor mental health outcomes among older parents who are left behind.” Is this true. There are several studies in Nepal and elsewhere on this issue. Please write a strong rationale for this study. What is the contribution of this study?

I would also expect a detailed review of the existing evidence of children’s migration and left-behind parent’s mental health, which is missing in the introduction.

The objective mentioned in line 78-80 and objectives section are not consistent. Pleas review.

Methodology:

Methodology or methods?

It is not clear how sampling process was implemented. Social security allowance in Nepal is provided to different age groups within the senior citizens based on ethnicity, widowhood and disability status. For example, older people for Brahmin and Chhetri ethnicity receive allowance at the age or 70 years (right?) and hence they are excluded from the sampling frame.

What were the questions you asked to ascertain empty nest? How many older people were reached out?

Sample size calculation: 8.18% would be 0.0818 in proportion, not 0.818. This makes the whole sample size calculation invalid.

What is judgmental sampling? On what factors did you make judgement for selecting qualitative samples? How did you ensure data saturation (line 133)?

Inclusion criteria: Please confirm that the inclusion criteria is participants who are living alone (line 144).

I am not sure about the conceptual framework you presented. It looks like a mediation or path analysis. Pleas revise.

Multivariate is different than multivariable. Similarly, bivariate??

Line 161-162: “Variables with a VIF of more than 2 and a p-value of more than 0.25 were excluded from the final model”. Please clarify p-value of what?

Line 164: Who are supervisor and co-supervisor? The manuscript looks like a thesis report.

What is abductive coding?

Ethics: How did you handle participants identified as having depression in the study?

Study measures and variables section in the methods section is completely missing.

Does GDS measure depression or depressive symptoms?

Results:

What does the numbers in parentheses in lines 196 to 200 mean?

In Table 2, please confirm <=2 categories for number of children and number of children abroad, as you already have 1 and 2 children.

How did you come up with depression level categories? What was the mean GDS score?

Did you use binary logistic regression or ordinal logistic regression? It is a bit confusing.

It looks like variables sex, occupation and household head were correlated. Then which one among the three did you retain in you model? I think you are not supposed to drop all of them.

Table 5 is missing.

Please write p < 0.001 instead of p = 0.000.

Qualitative results:

It looks like many of the codes (which probably was the result of the interview guideline) were not directly related to depression. I would suggest focusing on the objective of the study – depression and mental health.

6. PLOS authors have the option to publish the peer review history of their article (what does this mean?). If published, this will include your full peer review and any attached files.

**Do you want your identity to be public for this peer review?** For information about this choice, including consent withdrawal, please see our Privacy Policy.

Reviewer #1: No

Reviewer #2: **Yes:** Deependra Kaji Thapa

---

## [Decision Letter · Decision Letter 1]

26 Aug 2025

PMEN-D-24-00540R1

Depression among older parents’ and their associated factors after adult children’s migration: A mixed method study in an urban setting of Kathmandu Metropolitan City

PLOS Mental Health

Dear Dr. Shrestha,

Thank you for submitting your revised manuscript to PLOS Mental Health. After careful consideration of the reviewer reports, we feel that the paper has greatly improved however the reviewers still made some comments that we would like to offer you the chance to address. Therefore, we invite you to submit a revised version of the manuscript that addresses the points raised during the review process. You will be able to find all comments below.

We look forward to receiving your revised manuscript.

Kind regards,

Karli Montague-Cardoso

Staff Editor

PLOS Mental Health

Journal Requirements:

Additional Editor Comments (if provided):

Reviewers' comments:

Reviewer's Responses to Questions

**Comments to the Author**

1. If the authors have adequately addressed your comments raised in a previous round of review and you feel that this manuscript is now acceptable for publication, you may indicate that here to bypass the “Comments to the Author” section, enter your conflict of interest statement in the “Confidential to Editor” section, and submit your "Accept" recommendation.

Reviewer #1: (No Response)

Reviewer #2: (No Response)

2. Does this manuscript meet PLOS Mental Health’s publication criteria? Is the manuscript technically sound, and do the data support the conclusions? The manuscript must describe methodologically and ethically rigorous research with conclusions that are appropriately drawn based on the data presented.

Reviewer #1: Partly

Reviewer #2: Partly

3. Has the statistical analysis been performed appropriately and rigorously?

Reviewer #1: N/A

Reviewer #2: Yes

4. Have the authors made all data underlying the findings in their manuscript fully available (please refer to the Data Availability Statement at the start of the manuscript PDF file)?

Reviewer #1: No

Reviewer #2: Yes

5. Is the manuscript presented in an intelligible fashion and written in standard English?

Reviewer #1: Yes

Reviewer #2: No

6. Review Comments to the Author

Reviewer #1: I appreciate the authors' revision. While the manuscript has improved in clarity, the Introduction still does not adequately address the earlier comments. It remains too brief, it does not yet synthesize the relevant literature, and it is not anchored in a clear theoretical framework. I encourage the authors to expand this section so that it motivates the research questions, justifies the mixed-methods approach, and clarifies the study’s unique contribution.

1. A stronger literature review would situate parents’ mental health within the cultural realities of migration in the region. Rather than listing studies, please offer a concise synthesis that explains what we know and where findings diverge. For example, discuss how norms of filial piety, obligations around intergenerational support, and the social meaning of being a “migrant household” might increase or buffer depression risk among older parents. Consider the dual role of remittances and regular communication as potential protective resources, alongside the strain created by reduced proximal caregiving. It would help to specify what is actually limited in prior work to clarify how the present study advances the literature.

2. The introduction and discussion would also benefit from an explicit, culture-grounded theoretical framework and elaboration on the underlying mechanisms. Please explain that psychological outcomes around the “empty nest” reflect a balance between role loss and role-strain relief, and that this balance is shaped by culturally patterned factors such as familial roles and obligations and social norms about when and why children leave in collectivist culture. Discuss how remittances as moral obligations, frequency and quality of transnational contact, intergenerational co-residence, and community participation might exacerbate or buffer depression risk for parents with all children abroad. Articulating this culture-grounded framework will be useful for the readers to understand mixed findings in the literature and a clear rationale for your study. There is a lack of elaboration on the mechanisms underlying the studied relationship.

3. An adjusted OR of 28.3 for being single is extremely large. Consider to check for sparse cells, separation, influential cases, or mis-specification, and report model fit indices and diagnostic

4. The results section is improved. What would strengthen the mixed-methods claim is an explicit “joint display” that positions qualitative themes alongside the corresponding quantitative findings, especially around marital status/widowhood, living arrangement, communication, and material support, so readers can see where narratives explain the patterns.

5. The discussion should also expand more on the theoretical implications of the findings. This would help to further expand the discussion section.

6. For the qualitative component, if raw transcripts cannot be shared, consider depositing a de-identified excerpt set plus the codebook (with sensitive details masked) to meet transparency goals while protecting participants.

Reviewer #2: Thank you for addressing the earlier comments. The revised manuscript shows improvement compared to the initial version. However, I still have a few observations and suggestions for further clarification and refinement:

1. Please clarify the interpretation of the ordinal logistic regression findings. As currently written (in the abstract and results section), the statement “the odds of being likely to suffer from depressive symptoms was 28.3 times …” reads like a binary logistic regression interpretation. In ordinal logistic regression, the model estimates cumulative odds. Therefore, the result should be phrased as: single individuals had 28.3 times higher cumulative odds of being in a higher category of depressive symptoms (mild or moderate/severe vs. normal, and moderate/severe vs. mild/normal) compared to married individuals, holding other variables constant.

2. Did you check the proportional odds (parallel slopes) assumption of ordinal logistic regression? This assumption requires that the relationship between each pair of outcome groups is the same—that is, the effect of predictors on the odds of being in a higher versus lower category is constant across thresholds. Please report whether this assumption holds.

3. Please delete the phrase “As the sampling frame was unavailable for the above 60 population” (page 7, lines 111–112).

4. My understanding is that the social security allowance is provided to older adults above 60 years of age from certain groups (e.g., ethnic minorities, Dalits, widows), but not universally to all. For example, older adults from Brahmin/Chhetri groups are eligible only from age 70. If correct, this would mean the sampling frame is not representative of all older adults. Please clarify and, if applicable, acknowledge this as a limitation.

5. Please explain the rationale for selecting the age cut-off of 68 years, as well as the cut-off of 23 years for the length of the last child’s migration (Table 2).

6. If the study population consists of empty-nest older people, why is there a discrepancy between the number of children and the number of children abroad? Shouldn’t all children be out-migrated abroad? Please clarify.

7. The sentence “Since the results were reported as ‘Moderate/Severe and Mild versus Normal’ which basically is automatically divided into ‘depression’ vs ‘no depression’” (page 17, lines 236–238) is unclear and should be revised for clarity.

8. I recommend using the terms research team and subject experts instead of supervisor and co-supervisor.

9. I also suggest having the manuscript reviewed by someone experienced in editing and proofreading academic English to improve clarity and grammar.

7. PLOS authors have the option to publish the peer review history of their article (what does this mean?). If published, this will include your full peer review and any attached files.

**Do you want your identity to be public for this peer review?** For information about this choice, including consent withdrawal, please see our Privacy Policy.

Reviewer #1: No

Reviewer #2: **Yes:** Deependra Kaji Thapa, School of Public Health - Bloomington, Indiana University, USA

---

## [Decision Letter · Decision Letter 2]

9 Feb 2026

PMEN-D-24-00540R2

Depression among older parents’ and their associated factors after adult children’s migration: A mixed method study in an urban setting of Kathmandu Metropolitan City

PLOS Mental Health

Dear Dr. Shrestha,

Thank you for submitting your revised manuscript to PLOS Mental Health. As you will see from the reviewer comments below and attached, they are not completely satisfied with all of the changes and we would therefore like to offer you an additional round of revisions to rectify this. To prevent additional back-and-forth, please ensure that the remaining concerns are thoroughly addressed.

We look forward to receiving your revised manuscript.

Kind regards,

Karli Montague-Cardoso

Staff Editor

PLOS Mental Health

Journal Requirements:

Additional Editor Comments (if provided):

Reviewers' comments:

Reviewer's Responses to Questions

**Comments to the Author**

1. If the authors have adequately addressed your comments raised in a previous round of review and you feel that this manuscript is now acceptable for publication, you may indicate that here to bypass the “Comments to the Author” section, enter your conflict of interest statement in the “Confidential to Editor” section, and submit your "Accept" recommendation.

Reviewer #1: (No Response)

Reviewer #2: (No Response)

2. Does this manuscript meet PLOS Mental Health’s publication criteria? Is the manuscript technically sound, and do the data support the conclusions? The manuscript must describe methodologically and ethically rigorous research with conclusions that are appropriately drawn based on the data presented.

Reviewer #1: (No Response)

Reviewer #2: Yes

3. Has the statistical analysis been performed appropriately and rigorously?

Reviewer #1: (No Response)

Reviewer #2: No

4. Have the authors made all data underlying the findings in their manuscript fully available (please refer to the Data Availability Statement at the start of the manuscript PDF file)?

Reviewer #1: No

Reviewer #2: (No Response)

5. Is the manuscript presented in an intelligible fashion and written in standard English?

Reviewer #1: (No Response)

Reviewer #2: No

6. Review Comments to the Author

Reviewer #1: I reviewed the revised manuscript and rebuttal. The revision is improved overall, particularly in the Introduction’s attention to the cultural context of migration and caregiving, the clearer presentation of results, and the added effort to integrate quantitative and qualitative findings. However, there are several issues from my prior review remain only partial addressed:

1. The extremely large adjusted odds ratio for marital status (AOR ≈ 28.3) remains a substantive concern. Although the rebuttal indicates that diagnostic checks were conducted, these checks are not reported in the manuscript itself. Given the magnitude and wide confidence interval of this estimate, the manuscript should include supporting information such as a simple cross-tabulation of depression categories by marital status, clarification of any sparse cells, and at least one sensitivity analysis. That will help to improve transparency.

2. The added data triangulation section and joint display are a clear step forward and do help support the mixed-methods claim. At the moment, though, the joint display is doing most of its work around marital status. It would be even stronger if the authors could expand this table, or add a second one, to more clearly bring in living arrangements, communication patterns, and material support.

3. For the qualitative component, sharing the codebook is a helpful step toward transparency. That said, it would be useful if the Data Availability Statement were a bit clearer about why full transcripts cannot be shared and whether de-identified excerpts (beyond those already quoted in the paper) could be made available in some controlled way

4. The discussion has clearly been strengthened, especially the way it unpacks spousal loss and the idea of a “double loss” in empty-nest households. This interpretation comes across as thoughtful and is nicely supported by both the quantitative and qualitative results. It would be even more compelling if the authors could push this a bit further to reflect on the wider migration–care context. For example, why factors like communication or remittances did not emerge as independent predictors, despite featuring prominently in the narratives.

Reviewer #2: Thank you for providing your response to my comments. After reading it, a few additional concerns have arisen:

1. The interpretation of ordinal logistic regression still does not seem correct. In an ordered logit model with three outcome categories (as appears to be the case here), the proportional odds interpretation concerns the odds of being in a higher versus lower category across the cumulative cut-points, not a single binary comparison.

However, the output you provided raises a more fundamental concern:

How was the outcome ordered/coded in the ordinal model (i.e., the exact category order)? Based on the intercept labels, the ordering appears to be Mild → Moderate/Severe → No Depression, which is not the usual severity order (No Depression → Mild → Moderate/Severe). This can occur when outcome levels are left in alphabetical order rather than being explicitly set as an ordered factor. Because the meaning of a “higher category” depends entirely on the coding order, an incorrect order would make the reported coefficient/odds ratio interpretation misleading. Please re-check and clarify the coding/order used in the ordinal model.

2. The output you provided corresponds to the marital status coefficient for widowed/divorced/separated (β=−2.126) and an odds ratio of 0.119 (95% CI approximately 0.057–0.240). This is not compatible with the adjusted OR reported in the manuscript (28.3; 95% CI 9.19–103). Please confirm which model/output corresponds to the results reported in Table 6 and reconcile this discrepancy.

3. I did not understand what you mean by “The age cut-off of 68 years (during the time of data collection) was selected based on the eligibility threshold for receiving the Government of Nepal’s old-age social security allowance.” Is this true? Also, “The median age was found to be 23 years, hence it was selected.” Not sure whose median age are you referring to here as 23 years?

4. The Brant output contradicts your claim that “p-values for all variables are > 0.05” and that the assumption holds. You can see in the stata output you provided: Omnibus test: χ2=13.29, df = 1, p = 0 and Variable-specific test (marital status): χ2=13.29, df = 1, p = 0 (p < 0.001). This means the proportional odds assumption has not met.

5. When the software outputs p = 0.000, please report this as p < 0.001, as the probability is not literally zero.

6. “The multivariable ordinal logistic regression revealed only the variable marital status was the independent predictor of depression outcomes.” Not sure what you mean by ‘independent’ here?

7. PLOS authors have the option to publish the peer review history of their article (what does this mean?). If published, this will include your full peer review and any attached files.

**Do you want your identity to be public for this peer review?** For information about this choice, including consent withdrawal, please see our Privacy Policy.

Reviewer #1: No

Reviewer #2: **Yes:** Deependra K Thapa

Figure Resubmissions:

---

## [Decision Letter · Decision Letter 3]

6 Apr 2026

PMEN-D-24-00540R3

Depression among older parents’ and their associated factors after adult children’s migration: A mixed method study in an urban setting of Kathmandu Metropolitan City

PLOS Mental Health

Dear Dr. Shrestha,

Thank you for submitting your manuscript to PLOS Mental Health. After careful consideration, we would like to request one more round of revision to address the remaining, minor comments from the reviewer, which you can find below. Thank you for your understanding with this extended process.

We look forward to receiving your revised manuscript.

Kind regards,

Karli Montague-Cardoso

Staff Editor

PLOS Mental Health

Journal Requirements:

Additional Editor Comments (if provided):

Reviewers' comments:

Reviewer's Responses to Questions

**Comments to the Author**

1. If the authors have adequately addressed your comments raised in a previous round of review and you feel that this manuscript is now acceptable for publication, you may indicate that here to bypass the “Comments to the Author” section, enter your conflict of interest statement in the “Confidential to Editor” section, and submit your "Accept" recommendation.

Reviewer #2: (No Response)

2. Does this manuscript meet PLOS Mental Health’s publication criteria? Is the manuscript technically sound, and do the data support the conclusions? The manuscript must describe methodologically and ethically rigorous research with conclusions that are appropriately drawn based on the data presented.

Reviewer #2: (No Response)

3. Has the statistical analysis been performed appropriately and rigorously?

Reviewer #2: (No Response)

4. Have the authors made all data underlying the findings in their manuscript fully available (please refer to the Data Availability Statement at the start of the manuscript PDF file)?

Reviewer #2: (No Response)

5. Is the manuscript presented in an intelligible fashion and written in standard English?

Reviewer #2: (No Response)

6. Review Comments to the Author

Reviewer #2: Thank you for addressing the comments and providing a revised manuscript. Following are my additional comments:

1. Did the outputs/results of multivariable analysis in Table 6 change after adjusting the coding error of depression variable?

2. The title and footnotes of Table 6 should be revised to explicitly state that the estimates are derived from an ordinal (proportional odds) logistic regression model, as the current presentation may still be misinterpreted as binary logistic regression.

3. The adjusted odds ratio reported for family type (nuclear) appears inconsistent with the ordinal logistic regression output, as it does not reflect the exponentiated model coefficient and reverses the direction of effect; this estimate should be recalculated and corrected.

4. How did you calculate the bivariate (unadjusted) odds ratios in Table 6? They also need to be based on separate ordinal logistic regression models, this should be stated explicitly (e.g., using polr(depression ~ predictor, …..)), as unadjusted estimates from cross‑tabulations or binary models would not be directly comparable to the multivariable ordinal results.

5. Please correct any instances where p = 0.000 has been rewritten as p = 0.001; the appropriate convention is p < 0.001.

6. It looks like you used R for analysis, but you mentioned STATA?

7. Because you mentioned that there were wrong outputs attached at multiple instances, I highly recomment to upload the raw data and analysis code in teh supplementary file for transparency and reproducibility purpose. This would further ensure the integrity of this important research.

7. PLOS authors have the option to publish the peer review history of their article (what does this mean?). If published, this will include your full peer review and any attached files.

**Do you want your identity to be public for this peer review?** For information about this choice, including consent withdrawal, please see our Privacy Policy.

Reviewer #2: **Yes:** Deependra K Thapa

Figure Resubmissions:

---

## [Editor Report · Decision Letter 4]

10 Apr 2026

Depression among older parents’ and their associated factors after adult children’s migration: A mixed method study in an urban setting of Kathmandu Metropolitan City

PMEN-D-24-00540R4

Dear Ms Shrestha,

We are pleased to inform you that your manuscript 'Depression among older parents’ and their associated factors after adult children’s migration: A mixed method study in an urban setting of Kathmandu Metropolitan City' has been provisionally accepted for publication in PLOS Mental Health.

Best regards,

Karli Montague-Cardoso

Staff Editor

PLOS Mental Health